# OpenReview forum: "Risks from Language Models for Automated Mental Healthcare: Ethics and Structure for Implementation"
_colmweb.org/COLM/2024/Conference — COLM_

### Official Review · Reviewer_tFYA · 2024-05-08

**Rating:** 7
**Confidence:** 2
**Ethics Flag:** 1

**Summary:**

The paper presents some experiments to see whether LLM are ready to be used in TAIMH (Task-Autonomous AI in Mental Healthcare) systems.

**Questions To Authors:**

Would it be possible to test some fine tuning of the systems, and if so, how can it be performed?

**Reasons To Accept:**

Most of the paper is devoted to define what is a TAIMH system and this part is interesting to read and offers several references on this topic.

The experiments are based in a series of questionnaires fed into several LLMs to analyze their output. The evaluation has been performed manually labeling it by at least two psychiatrists independently. The evaluation has been performed over quite a lot of systems.

**Reasons To Reject:**

The improvements of the systems are based on improving the systems prompts but no fine-tuning of the systems have been performed.

---

> ### Author Rebuttal · Authors · 2024-05-31
>
> Dear reviewer,
>
> We thank you for your review, and we genuinely appreciate the constructive tone and helpful feedback! We are excited that you see the potential importance of our work.
>
> We agree with your suggestion that the impact of fine-tuning LLMs for these mental health applications (either in general or to improve model safety) would enhance our study. To this end, we evaluated Meditron-7B and Meditron-70B [1, 2] this week, are currently evaluating MentaLLaMA [3], and are planning to evaluate PMC-LLaMA [4], too. When all results are collected, we will use the added page for the final paper to add them as an additional table to showcase how task-specific fine-tuning affects the general safety performance (as tested in Table 2, Section 4.2) and how the tested safety improvement approaches in Section 5.2 are affected.
>
> If (mental) health fine-tuned models are better at self-critiques of unsafe responses, we can claim that the Constitutional AI and RLAIF framework [5] could be used to enhance safety through fine-tuning at scale. However, without a functioning self-critique loop (or similar), mental health-specific safety fine-tuning would require a new and larger red-teaming data set to use, e.g., the labeled samples of safe and unsafe responses for Direct-Preference-Optimization (DPO) [6]. However, for DPO, our collected data set of 16 questionnaires is not large enough to do fine-tuning and statistically robust evaluations afterward.
>
> We thank the reviewer for the suggestion and will expand the paper to include these results together with an extended discussion.
>
> Thank you so much for your time and considering our work for CoLM 2024,
> The authors
>
> References:
> [1] Chen et al. MEDITRON-70B: Scaling Medical Pretraining for Large Language Models. arxiv:2311.16079
> [2] https://ai.meta.com/blog/llama-2-3-meditron-yale-medicine-epfl-open-source-llm/
> [3] Yang et al. MentaLLaMA: Interpretable Mental Health Analysis on Social Media with Large Language Models. arxiv:2309.13567
> [4] PMC-LLaMA: Towards Building Open-source Language Models for Medicine
> [5] Bai et al. Constitutional AI: Harmlessness from AI Feedback. arxiv:2212.08073
> [6] Rafailov et al. Direct Preference Optimization: Your Language Model is Secretly a Reward Model. arxiv:2305.18290

---

> > ### Comment · Reviewer_tFYA · 2024-06-04
> >
> > Thank you for the answer.

---

### Official Review · Reviewer_UZ7w · 2024-05-10

**Rating:** 7
**Confidence:** 4
**Ethics Flag:** 1

**Summary:**

The authors have examined the readiness of language models' for providing mental healthcare - prompting 10 LLMs using prompts/ vignettes developed and evalauted with psychiatrists.

**Questions To Authors:**

1. Pg 5 - Please define "legibility" and "corrigibility"

2. Sec 5.2 - It is interesting that Approach-1 and 2 did not improve response for mania. Unsafe responses also increased on adding TAIMH values for psychosis. Could you elaborate on how responses changed (or, remain unchanged) compared to the responses obtained from original prompts? What ethical principles are violated? It might be useful to discuss what a safe/ethical response looks like.

3. Pg 2 - There is no previous work on how AI may augment the mental healthcare system...
In the Discussion, the authors mention "first of its kind framework" (Fig 1) - similar suggestions have been made in the past. Please see below works:

Stade, Elizabeth C., et al. "Large language models could change the future of behavioral healthcare: a proposal for responsible development and evaluation." npj Mental Health Research 3.1 (2024): 12.
Stade, E. C., Jane P. Kim, and Shannon Wiltsey Stirman. "Readiness Evaluation for AI Deployment and Implementation for Mental Health: A Review and Framework." (2024).

**Reasons To Accept:**

I appreciate actionable references (such as NICE) that can be useful to CS x Mental Health community for designing relevant capabilities. The findings emphasize the need to examine language models for behavioral health outcomes and safety of vulnerable users.

**Reasons To Reject:**

Please see Questions to Authors

---

> ### Author Rebuttal · Authors · 2024-05-31
>
> Dear reviewer,
>
> We thank you for your review, and we genuinely appreciate the constructive tone and helpful feedback! We are excited that you see the potential importance of our work.
>
> Question 1:
>
> In the context of TAIMH, we think of "legibility" and "corrigibility" as follows: Legibility refers to the AI system's ability to make its decision-making processes and recommendations clear and understandable to both patients and healthcare providers. Corrigibility means that the AI system can accept and effectively implement corrections or adjustments from healthcare professionals to ensure patient safety and alignment with therapeutic goals. Together, these attributes ensure that the AI is both transparent and adaptable, fostering trust and enhancing the quality of care.
>
> We definitely should have better defined these terms in the text, and we will explain them in more detail in the final version. We thank the reviewer for pointing out this ambiguity!
>
> Question 2:
>
> When evaluating whether different approaches improve the safety of LLMs in Section 5.2, we reviewed the responses identically as for the experiments in Section 5.1 (Table 2), as defined in the rubric in Appendix C. Essentially, safe and unsafe responses for Psychosis and Mania did not change in any meaningful way, only the frequency of safe and unsafe responses. We list example answers of safe and unsafe responses for each question in Appendix D.
>
> Interestingly, for most tested models, the safe and unsafe responses can be as different as night and day for the same model and prompt. The difference between safe and unsafe responses was less subtle than we initially thought. We will add more details to the final version of the main paper to better communicate these nuances.
>
> Question 3:
>
> We thank the reviewer for suggesting these works and will gladly add them to the related work section! To better position our work, we will highlight the differences between our framework and the proposed ones in these references.
>
> Thank you so much for your time and considering our work for CoLM 2024,
> The authors

---

### Official Review · Reviewer_267G · 2024-05-11

**Rating:** 7
**Confidence:** 3
**Ethics Flag:** 1

**Summary:**

In this paper, the authors propose a framework to evaluate whether a given TAIMH (task-autonomous AI in mental health care) is suited to be deployed in psychiatric emergencies and follows the ethical requirements needed in this health care area. Thus, the authors proposed a list of requirements for any TAIMH, including imperatives like ‘discourage and prevent harm to users and others’. Ten large language models were tested in five psychiatric emergencies covering depression, self-harm, psychosis, mania, suicidal and homicidal thoughts. The tests were guided by psychiatrists. Results show that no tested model is ready for TAIMH. Then the authors applied modifications in the prompts given to the LLM in order to test if the LLM would give safer answers; this last test was done only on psychosis and mania and only one family of LLM. The results of this last test show that other methodologies are needed to prepare LLM for TAIMH. The work is relevant to the current state of use of LLM for health care. The paper is well-written and easy to understand.

**Reasons To Accept:**

* The paper is relevant given the impact of large language models being used in mental health, with real consequences of undesired use of this technology.
* An ethical framework and a way to evaluate how safe the LLM's output is for direct use by people in psychiatric emergencies is needed. This paper provides the first steps towards both aspects.
* The design of the test is done by health professionals, and so it is the evaluation of answers.

**Reasons To Reject:**

* It is not clear to me how others can reproduce the results. The LLM used is not described in detail.

---

> ### Author Rebuttal · Authors · 2024-05-31
>
> Dear reviewer,
>
> We thank you for your review, and we genuinely appreciate the constructive tone and helpful feedback! We are excited that you see the potential importance of our work.
>
> We appreciate the concern that our results might be hard to replicate, presumably given that mental health clinicians (M.D.s) evaluated model responses. In the submitted paper, we have an expert-created rubric in Appendix C that defines requirements for each response classification for each question as well as example answers of safe and unsafe responses for each question in Appendix D. Combined, the rubric, and labeled examples offer non-clinicians the ability to replicate the results generally and should enable perfect replication when working with mental health clinicians.
>
> We are unsure if the reviewer feedback might also refer to the clarity of section 5.2 (Improving Safety), but we will use the additional page for the final version to explain the experimental setup and results in more detail.
>
> Thank you so much for your time and considering our work for CoLM 2024,
> The authors

---

> > ### Comment · Reviewer_267G · 2024-06-06
> >
> > I have read the author's responses. Thank you.
> > I believe the additional information would improve the paper. And I would like to see this paper accepted.

---

### Official Review · Reviewer_kF2r · 2024-05-12

**Rating:** 6
**Confidence:** 5
**Ethics Flag:** 1

**Summary:**

The paper proposes a definition and framework for Task-Autonomous AI in Mental Health (TAIMH), focusing on ethical considerations with the potential in augmenting mental healthcare systems with varying levels of autonomy and intervention

The study evaluates ten SOTA language models using questionnaires designed to assess their ability to detect and manage basic psychiatric symptoms. The study focuses on the Llama-2 family of models while its focus within the questionnaires lies on those instances where nearly all models generated unsafe responses, particularly in the areas of Psychosis and Mania.
Psychiatrists conducted the evaluations and found that none of the models matched human professionals in nuanced understanding and contextual appreciation. Most models displayed issues such as overly cautious or sycophantic responses and lacked necessary safeguards, raising concerns about potential harm if accessed by users in mental health crises.

The paper concludes by suggesting improvements for existing models to enhance user protection and facilitate future TAIMH applications.

**Questions To Authors:**

The models you have used are not domain adapted for mental health; useful to reconsider your evaluation results.

**Reasons To Accept:**

A very useful evaluation of the use of AI for mental healthcare systems
The use of human experts

**Reasons To Reject:**

The study should consider models which have been tailored to mental health such as
https://arxiv.org/pdf/2309.13567. Thus, the conclusions need to be re-assessed.

---

> ### Author Rebuttal · Authors · 2024-05-31
>
> Dear reviewer,
>
> We thank you for your review, and we genuinely appreciate the constructive tone and helpful feedback! We are excited that you see the potential importance of our work.
>
> We agree with your suggestion that testing LLMs fine-tuned for (mental) health applications would enhance our studies. To this end, we evaluated Meditron-7B and Meditron-70B [1, 2], are currently evaluating the reviewer-proposed MentaLLaMA [3], and are planning to evaluate PMC-LLaMA [4]. When all results are collected, we will use the added page for the final paper to add them as an additional table.
>
> The results for Meditron-7B and Meditron-70B show significantly more unsafe responses to users in mental health emergencies across all questions. Qualitatively, the models seem to try to add medical details to harmful user queries, e.g., from the suicide category, instead of giving safe responses. We thank the reviewer for the suggestion and are excited to see how MentaLLaMa and PMC-LLaMA will perform.
>
> We would also like to explain why we focused on general state-of-the-art LLMs. While we expect models fine-tuned on mental health information to perform well clinically, we thought it was more vital to investigate the safety of models readily accessible to all users with safety and alignment to human preferences at the core of their development. Given their vast user numbers, these models must recognize and triage psychiatric emergencies appropriately. The likelihood that an individual seeks out a model while in crisis increases with the ubiquity of the technology, and there has already been a reported suicide directly related to a model's inability to detect and triage suicidality appropriately. Even though Google Search is not a fine-tuned medical resource, it still must robustly detect and respond to psychiatric emergencies. We view universal tools like ChatGPT in a similar way, and we believe such tools should be held to the same (if not a higher) standard.
>
> Thank you so much for your time and considering our work for CoLM 2024,
> The authors
>
> Ref:
> [1] Chen et al. MEDITRON-70B: Scaling Medical Pretraining for Large Language Models. arxiv:2311.16079
> [2] https://ai.meta.com/blog/llama-2-3-meditron-yale-medicine-epfl-open-source-llm/
> [3] Yang et al. MentaLLaMA: Interpretable Mental Health Analysis on Social Media with Large Language Models. arxiv:2309.13567
> [4] Wu et al. PMC-LLaMA: Towards Building Open-source Language Models for Medicine. arxiv:2304.14454

---

> > ### Comment · Reviewer_kF2r · 2024-06-04
> > **Response**
> >
> > The authors have addressed all my questions and recommendations. I am happy to see this paper accepted to the conference

---

### Decision · Program_Chairs · 2024-07-10

**Decision:**

Accept

**Comment:**

This paper makes a valuable and timely contribution by proposing a framework to evaluate the safety and ethics of language models for task-autonomous AI in mental healthcare. The authors define clear requirements, systematically test state-of-the-art LLMs against clinically-relevant scenarios, and highlight current deficiencies in model safety. The reviewers have reached a unanimous decision to accept the paper.